# Single catheter strategy for transradial angiography and primary percutaneous coronary intervention enhances procedural efficiency, microvascular outcomes, and cost-effectiveness: Implications for STEMI healthcare in resource-limited settings

Mohajit Arneja[1☯], Swetharajan Gunasekar[2☯], Dharaneswari Hari Narayanan[2☯], Joshma Joseph[1], Harilalith Kovvuri[1], Sharath Shanmugam[1], Pavitraa Saravana Kumar[2], Asuwin Anandaram[3], Vinod Kumar Balakrishnan[1], Jayanty Venkata Balasubramaniyan[1], Sadhanandham Shanmugasundaram[1], Sankaran Ramesh[1], Nagendra Boopathy Senguttuvan[1]*

1 Department of Cardiology, Sri Ramachandra Institute of Higher Education and Research (SRIHER), Chennai, Tamil Nadu, India, 2 Department of Clinical Research, Sri Ramachandra Institute of Higher Education and Research (SRIHER), Chennai, Tamil Nadu, India, 3 Department of Internal Medicine, Lehigh Valley Health Network, Allentown, Pennsylvania, United States of America

☯ These authors contributed equally to this work.
* drsnboopathy@gmail.com

## Abstract

### Background

Faster time to reperfusion can be achieved by minimizing various patient and system-level delays that contribute to total ischemic time. Procedural delays within the catheterization laboratory represent a non-negligible and modifiable component in the chain of reperfusion, but remain unquantified by conventional metrics such as door-to-ballon (D2B) time. Universal catheter approaches have rapidly gained traction as an alternative to the traditional two catheter approach for transradial coronary interventions. However, their utility for both diagnostic angiography and subsequent angioplasty is limited, and the impact of this strategy on reperfusion outcomes has remained unexplored. We utilized a procedural metric termed fluoroscopy-to-device (FluTD) time to quantify the efficiency of a single catheter strategy, and assessed its impact on epicardial and myocardial perfusion.

### Methods and results

In this retrospective study, consecutive STEMI patients undergoing transradial primary PCI (pPCI) at a tertiary care center in India between May 2022 to October 2024 were analyzed. Patients were divided into two groups: 51 underwent PCI

**Data availability statement:** All relevant data are within the manuscript and its Supporting Information files.

**Funding:** The author(s) received no specific funding for this work.

**Competing interests:** The authors have declared that no competing interests exist.

using a single universal guiding catheter (UGC), and 51 underwent the conventional two-catheter (CTC) approach. The primary outcome of the study was a comparison of the FluTD time between the two procedural strategies. Secondary outcomes included myocardial blush grade (MBG), Thrombolysis in Myocardial Infarction (TIMI) flow grade, total fluoroscopy time, radiation dose, device safety and efficacy, and procedural success.

The median FluTD time was significantly shorter in the UGC compared to the CTC group (3 minutes [IQR 3–4] vs. 10 minutes [IQR 8–17], p < 0.001), with a higher proportion of patients in the former achieving myocardial blush grade (MBG) of 3 (86.3% vs. 54.9%, p = 0.004), indicating superior microvascular reperfusion. Despite a higher incidence of bifurcation lesions (33.3% vs 11.8, p = 0.04) and left main (LM) interventions (7.8% vs 0%, p = 0.04) among patients in the UGC cohort, the single catheter strategy maintained superior procedural efficiency without increased complication rates.

## Conclusion

A single catheter strategy for both angiography and pPCI in STEMI patients was associated with a significant reduction in FluTD time and improved microvascular perfusion, without compromising device safety or efficacy. In low- and middle-income countries (LMICs), where intra- and extra-procedural delays are often more pronounced, inclusion of the single catheter strategy can optimize catheterization workflows and yield substantial cost-savings.

## Introduction

Timely reperfusion is critical to the restoration of epicardial flow and salvaging the myocardium in patients presenting with ST-segment elevation myocardial infarction (STEMI). Primary percutaneous coronary intervention (pPCI) remains the dominant reperfusion strategy and produces superior mortality and reinfarction outcomes compared to fibrinolytic therapy [1–4]. This advantage is contingent upon prompt initiation of percutaneous coronary intervention (PCI) from the time of onset of symptoms to the time of first medical contact (FMC). Current guidelines recommend PCI for all patients who present within 12 hours of symptom onset, and further advocate that device engagement leading to re-establishment of flow in the culprit vessel be accomplished within 90 minutes from the time of patient arrival at a PCI-capable site [5–7]. This 90-minute window is often referred to as the door-to-balloon (D2B) time, and shorter D2Bs have been consistently associated with reduced short and long-term mortality risk in large, patient-level observational studies [8–12].

Although D2B time remains one among many important variables in setting the standard for achieving favorable PCI outcomes, its validity as a performance metric is largely limited to countries with robust pre-hospital infrastructure. In addition, D2B time does not isolate and quantify procedural delays that are inherent to

catheterization laboratory workflows and choice of PCI technique. The latter warrants further study, as an optimal procedural strategy combined with the appropriate interventional device, may be critical in reducing time-to-reperfusion. Towards this end, existing studies in the literature have proposed the use of a single catheter for diagnostic angiography to optimize various intraprocedural parameters, but their role in streamlining subsequent PCI has received limited attention. Innovative catheter designs such as the Ikari Left (IL) guiding catheter (Heartrail III, Terumo Corporation, Tokyo, Japan) facilitate universal engagement of the left and right coronary arteries and enable simultaneous angiography and PCI, thereby minimizing or bypassing the requirement for a separate diagnostic catheter and an additional interventional catheter [13–15]. While this single catheter-based angiography and PCI technique has been associated with significant reductions in D2B time [16–18], the direct clinical implication(s) of this time reduction remains elusive. Myocardial perfusion is one such outcome that remains underexplored within the context of procedural optimization. It merits consideration given the wealth of data that suggests that, although contemporary PCI has enabled successful restoration of epicardial patency, suboptimal microcirculatory perfusion is evident in almost 50% of patients [19,20]. Despite this, no studies have evaluated whether enhanced procedural efficiency translates into improved cardiovascular outcomes. In this real-world, retrospective pilot study, we established a procedural metric to quantify the efficiency of single catheter-based angiography and PCI, and evaluated its impact on epicardial and myocardial outcomes.

## Methods

### Study design

This single-center, retrospective observational study was conducted at the Department of Cardiology at Sri Ramachandra Institute of Higher Education and Research (SRIHER), India, from May 2022 to October 2024. The study was approved by the Institutional Ethics Committee (IEC) at Sri Ramachandra Institute of Higher Education and Research (SRIHER), India (Ref: CSP-MED/25/JAN/112/02). The need for informed consent was waived due to the retrospective nature of the study. Consecutive patients aged 18 years and older who underwent transradial pPCI for STEMI during this period were identified through the institutional cardiac catheterization database. Patients were excluded if they had undergone transfemoral interventions or were pregnant. During the preliminary stage of data collection, the authors had access to personal identifiers such as patient name, hospital identification number (ID), age, and sex as these parameters are routinely reported on coronary angiograms, PCI reports, and discharge summaries. Highly specific personal identifiers such as patient names and IDs were subsequently removed, and anonymized data was extracted from the database, including baseline demographic details, clinical characteristics, angiographic and interventional findings, as well as procedural metrics. Timestamps for procedural steps were collected from three state-of-the-art cardiac catheterization laboratories: Philips Azurion 7M12, GE G-XL-104313, and Philips Allura Xper FD 20/10. The study adhered to the Strengthening the Reporting of Observational Studies in Epidemiology (STROBE) guidelines for facilitating ascertainment and critical appraisal of the study design, methodology, and findings [21].

### Sample size calculation

Sample size calculations were performed using ClinCalc.com. It was assumed that the mean (SD) time would be 8 (4) minutes in the conventional two catheter group and 5 minutes in the UGC group. To detect this difference with a power of 90% and a Type I error rate ($\alpha$) of 0.05, 37 patients were needed in each arm (total = 74) under a 1:1 enrollment ratio. Anticipating a 10% rate of non-interpretable angiograms and a 10% crossover to femoral access, a total sample size of 90 patients (45 in each group) was estimated to ensure detection of the desired difference.

### Patient grouping and operator experience

To compare procedural strategies, patients were grouped into two cohorts. The first cohort of patients underwent both angiography and PCI using a single, universal guiding catheter (UGC), whereas separate diagnostic and guiding catheters

were used for patients in the conventional two-catheter (CTC) cohort. All eligible, consecutive STEMI patients in whom pPCI was performed with the IL guiding catheter during the study period (May 2022-October 2024) were first identified via the institutional catheterization database (n = 51) and included in the study. Working retrospectively from October 2024, we then included the first 51 consecutive STEMI patients who were treated using the two-catheter approach, until the adequate sample size was reached, while maintaining equal allocation. Since the utilization of two catheters was more common and practiced by multiple operators, we reached the sample size for the CTC group (n = 51) over an earlier timeframe (September 2023-October 2024). Despite asymmetric timelines, institutional protocols and transradial practices remained unchanged over the study period, and no further matching was performed. All investigators involved in the study had over 10 years of experience performing pPCI using the CTC strategy, while one investigator had 9 years of experience performing single catheter-based PCI. In the CTC cohort, initial diagnostic angiography was performed using the Tiger catheter (Optitorque®, Terumo Corporation, Tokyo, Japan). Later, a Judkins Right (JR) was used to access the right coronary artery (RCA), while a Judkins Left (JL) or an extra backup catheter was used for left coronary artery (LCA) interventions.

## Fluoroscopy-to-Device (FluTD) time measurement

Given the retrospective nature of the study, exact D2B times were not consistently available for all patients. To minimize biases arising from systemic or patient-related delays, such as time spent in the emergency department, cath lab activation, radial access complications, or challenges posed by subclavian artery tortuosity, a metric termed 'FluTD time' was utilized. FluTD time was defined as the time from the visualization of the guiding catheter in the ascending aorta to the entry of the coronary guidewire into the culprit vessel, providing a focused measure of procedural efficiency by excluding other factors that might skew overall D2B measurements.

## Catheterization technique for the Ikari Left guiding catheter

For the IL catheter, a 0.035″ J Tip Teflon Wire (Dr. Surgical, India) was used for introduction into the ascending aorta. The time of visualization of the guide catheter inside the ascending aorta was recorded as the first FluTD time. The non-culprit vessel was imaged first, with the 0.035" Teflon wire remaining in place to provide additional support and curve manipulation. Maintaining the 0.035" Teflon wire within the secondary curve helped the operator engage the RCA easily, while positioning it within the tertiary curve facilitated easier engagement of the LCA. A routine process of making the system air-free by back-bleeding was performed to minimize the risk of air embolism. In the CTC system, operators engaged in accordance with the standard of care, and in this strategy, the diagnostic catheter's visualization time in the ascending aorta was recorded as the starting time for FluTD.

## Study outcomes and definitions

The primary outcome of the study was to compare the FluTD time difference between the UGC and CTC strategies. Secondary outcomes included procedural success, total fluoroscopy time, total radiation dose, Thrombolysis in Myocardial Infarction (TIMI) flow grade and Myocardial Blush Grade (MBG) at the end of the procedure, device efficacy, device safety, engagement rate, change of access site, change of guide catheters, incidence of guide catheter-induced dissection, use of the mother-and-child technique for device delivery, and procedural complications. Device efficacy was defined as the ability of the device to provide stable engagement and efficient guiding support for intervening the culprit vessel. Device safety was defined as successful engagement without causing procedural complications such as ostial dissections. Procedural success was defined as achieving a minimum stenosis diameter reduction to <30% after balloon angioplasty and <10% after stenting, without major procedural complications such as in-hospital mortality, stent thrombosis, coronary artery perforation, or recurrent myocardial infarction (re-MI). Engagement rate was defined as the rate of successful engagement in the coronary ostium with the initially selected guiding catheter.

Flluoroscopy time was defined as the total fluoroscopy time recorded during the procedure, measured to the nearest 0.1-minute, based on the NCDR® CathPCI Registry® v4.4 Coder's data dictionary definitions [22]. Total radiation dose was defined as the total amount of radiation absorbed by the patient during the procedure, measured as cumulative air kerma in milligray (mgy) [23,24].

Change of access referred to modifications or adjustments in the vascular access site (e.g., switching from radial to femoral access) during the procedure. Change of guide catheter was defined as documented cases where the guiding catheter initially chosen for procedural access was switched to one better suited for successfully engaging and treating the culprit coronary vessel. Guide catheter-induced dissection was defined as any dissection event caused by the guiding catheter during engagement/course of intervention [25]. Use of the mother-and-child technique for device delivery was defined as the employment of a specialized technique involving an extension catheter (the "child") inserted through the main guiding catheter (the "mother") to enhance backup support for device delivery [26,27]. Post-procedural blood flow in the treated artery was assessed using the TIMI coronary flow grade, with 0 representing no flow and 3 signifying adequate restoration of epicardial flow [28]. Myocardial perfusion post-PCI was assessed angiographically using the Myocardial Blush Grade (MBG), with scores ranging from from 0 (no blush) to 3 (normal blush) [29,30]. Procedural complications included any adverse events such as in-hospital mortality, stent thrombosis, coronary artery perforation, or re-MI during the index hospitalization.

### Statistical analysis

Continuous variables were expressed as either the mean (SD) or median [IQR], depending on the normality and distribution of the data, as assessed using the Shapiro-Wilk test. Group comparisons employed parametric (Student's t-test) or non-parametric (Mann-Whitney U) tests, as appropriate. Categorical variables were represented as percentages and analyzed using the Chi-square test or Fisher's exact test. A log-linear regression model assessed independent predictors of FluTD time, with beta coefficients (β) representing percentage change in FluTD time (positive β = increased time; negative β = reduced time). Statistical significance was defined as a two-tailed p-value < 0.05. All statistical analyses were performed using SPSS version 27 (IBM, New York) and STATA 18 (StataCorp LLC, Texas, USA).

## Results

### Clinical presentation and preoperative profiles

The study included 102 patients, equally divided between the UGC (n = 51) and CTC (n = 51) cohorts. The baseline clinical characteristics of patients are summarized in Table 1. The median age of the participants was 55 years [IQR: 48–64], with no statistically significant difference between the groups (p = 0.446). The population was predominantly male (87.3%), with no significant gender differences (p = 0.554). Frequently encountered comorbidities such as diabetes mellitus (56.86%), hypertension (44.12%), and dyslipidemia (84.3%) were distributed similarly between the groups. Anterior wall myocardial infarction (MI) (56.9%) was the most common presentation, followed by inferior (40.2%) and posterior wall (2.9%) MI. Pre-procedural hemodynamic stability differed significantly between the cohorts, with the CTC group harboring a greater proportion of Killip Class I patients (66.7% vs. 39.2%, p = 0.01), while the UGC group had higher numbers of Killip Class II and IV presentations. The median left ventricular ejection fraction (LVEF) across both cohorts was 43% [IQR: 39–50], with no significant difference between the groups (p = 0.227) (Table 1).

### Angiographic and interventional findings

Coronary angiographic details and post-procedural outcomes of the study cohort are summarized in Table 2. Across both cohorts, the left anterior descending (LAD) artery was the most common culprit vessel, followed by the RCA, and the left circumflex artery (LCX). No significant differences were observed between the groups in terms of culprit vessel

**Table 1. Baseline clinical characteristics and pre-operative profiles of patients in the UGC and CTC cohorts.**

| | UGC (n = 51) | CTC (n = 51) | p-value |
|---|---|---|---|
| | N (%) or Median [IQR] | N (%) or Median [IQR] | |
| **Age** | 55 [44 −64] | 55 [49 −65 ] | 0.446 |
| **Sex** | | | |
| **Male** | 46 (90.2) | 43 (84.3) | 0.554 |
| **Female** | 5 (9.5) | 8 (15.7) | |
| **Comorbid Conditions** | | | |
| **DM** | 30 (58.8) | 28 (54.9) | 0.689 |
| **IDDM** | 4 (7.84) | 7 (13.73) | 0.338 |
| **Hypertension** | 22 (43.14) | 23 (45.1) | 0.841 |
| **Thyroid Disorders** | 0 (0) | 1 (1.96) | 0.314 |
| **Dyslipidemia** | 41 (80.4) | 45 (88.2) | 0.276 |
| **Previous CAD** | 1 (0.98) | 1 (1.96) | 0.314 |
| **CKD** | 5 (9.8) | 6 (11.7) | 0.740 |
| **PAD** | 1 (1.96) | 0 (0) | 0.315 |
| **Type of MI** | | | |
| **AWMI** | 28 (54.90) | 30 (59.00) | 0.060 |
| **IWMI** | 20 (39.00) | 21 (41.00) | |
| **PWMI** | 3 (5.90) | 0 (0.00) | |
| **Killip Classification** | | | |
| **I** | 20 (39.22) | 34 (66.7) | **0.010*** |
| **II** | 21 (41.18) | 14 (27.4) | |
| **III** | 0 (0) | 0 (0) | |
| **IV** | 10 (19.61) | 3 (5.8) | |
| **LVEF** | 43 [35 − 48] | 43 [40 − 50] | 0.227 |

Abbreviations: DM: Diabetes mellitus; IDDM: Insulin-dependent diabetes mellitus; CAD: Coronary artery disease; MI: Myocardial infarction; AWMI: Anterior wall myocardial infarction; IWMI: Inferior wall myocardial infarction; PWMI: Posterior wall myocardial infarction; CKD: Chronic kidney disease; PAD: Peripheral artery disease; Killip Class: Classification system for heart failure in myocardial infarction; LVEF: Left ventricular ejection fraction, SD: Standard Deviation, *Statistically significant, p-value < 0.05

involvement (p = 0.909). The UGC group exhibited greater lesion complexity, with higher rates of bifurcation lesions (33.3% vs. 11.8%, p = 0.009) and exclusive left main interventions (7.8% vs. 0%, p = 0.04). Other lesion characteristics, such as calcified and thrombotic lesions, did not differ significantly between the two groups. With respect to the lesion segments, the median length of drug-eluting stents (DES) used did not differ significantly between the two groups (28 mm vs 28 mm, p = 0.631). There were no significant differences in the final TIMI flow, with grade 3 TIMI flow achieved in 100% of patients in the single catheter-PCI cohort and in 96.1% of patients in the conventional group (p = 0.495). However, a significantly greater proportion of patients in the UGC cohort attained MBG 3 compared to the conventional group (86.3% vs 54.9%, p = 0.004), indicating satisfactory myocardial perfusion post-PCI (**Table 2**).

**Fluoroscopy-to-device time assessment**

Procedural efficiency as measured by FluTD time, and other related intra-procedural parameters such as radiation dose and fluoroscopy time are reported in **Table 3**. FluTD time was significantly shorter in patients who underwent single catheter-based angiography and PCI, with a median of 3 minutes [IQR: 3–4], compared to 10 minutes [IQR: 8–17] in patients who underwent PCI using the conventional two-catheter system (p < 0.0001). Although total radiation dose and fluoroscopy time were lower in the UGC cohort, these differences did not reach statistical significance (**Table 3**).

**Table 2. Procedural and angiographic outcomes in STEMI patients undergoing PCI with single (UGC) versus dual catheter (CTC) approaches.**

| | UGC (n = 51) | CTC (n = 51) | p-value |
|---|---|---|---|
| | N (%) or Median [IQR] | N (%) or Median [IQR] | |
| **Culprit Vessel** | | | |
| **LAD** | 28 (55) | 30 (58.8) | 0.909 |
| **RCA** | 17 (33.3) | 16 (31.4) | |
| **LCX** | 6 (11.7) | 5 (9.8) | |
| **Vessel Involvement** | | | |
| **Single Vessel Disease** | 27 (53) | 35 (68.7) | 0.178 |
| **Double Vessel Disease** | 13 (25) | 11 (21.5) | |
| **Triple Vessel Disease** | 11 (22) | 5 (9.8) | |
| **Pre-Procedure TIMI Flow** | | | |
| **0** | 24 (47) | 24 (47) | **0.010*** |
| **1** | 10 (20) | 14 (27) | |
| **2** | 17 (33) | 7 (13.7) | |
| **3** | 0 (0) | 6 (11.7) | |
| **Other Vessel Characteristics** | | | |
| **Bifurcation Lesion** | 17 (33.3) | 6 (11.8) | **0.009*** |
| **Calcified Lesion** | 4 (7.8) | 7 (13.7) | 0.338 |
| **Thrombotic Lesion** | 40 (78.4) | 41 (80.4) | 0.806 |
| **Left Main Disease** | 5 (9.9) | 2 (3.9) | 0.240 |
| **Left Main Intervention** | 4 (7.8) | 0 (0) | **0.040*** |
| **Pre-Dilatation** | 39 (76.5) | 37 (72.5) | 0.649 |
| **Post-Dilatation** | 49 (96.1) | 46 (90.2) | 0.240 |
| **Thrombosuction** | 12 (23.5) | 6 (11.7) | 0.119 |
| **Total DES Length (mm)** | 28 [23 - 41] | 28 [24 − 40] | 0.631 |
| **Post-Procedure TIMI Flow** | | | |
| **0** | 0 (0) | 0 (0) | 0.495 |
| **1** | 0 (0) | 0 (0) | |
| **2** | 0 (0) | 2 (3.92) | |
| **3** | 51 (100) | 49 (96.08) | |
| **Myocardial Blush Grade** | | | |
| **0** | 0 (0) | 5 (9.8) | **0.004*** |
| **1** | 1 (2.0) | 4 (7.8) | |
| **2** | 6 (11.8) | 14 (27.4) | |
| **3** | 44 (86.3) | 28 (54.9) | |

Abbreviations: LAD: Left anterior descending, LCX: Left circumflex, RCA: Right coronary artery, TIMI: Thrombolysis in Myocardial Infarction, MBG: Myocardial Blush Grade, DES: Drug-eluting Stent, IQR: Interquartile Range, * Statistically significant, p < 0.05.

## Interventional outcome measures

The results of other secondary outcome measures, including device efficacy andsafety, as well as procedural success and complication rates, are summarized in **Table 4**. The IL guiding catheter demonstrated high efficacy and stability, providing successful engagement in 50 out of 51 cases (98.04%). In one case involving a tortuous LCX, the guide catheter had to be switched for an extra backup catheter to achieve stable engagement. Procedural success was achieved in all patients across both cohorts. In terms of device safety, the IL catheter registered a safety rate of 98.04%, owing

**Table 3. Comparison of Fluoroscopy-to-device (FluTD) time, total fluoroscopy time, and total radiation dose between the UGC and CTC procedural strategies.**

| Intra-procedural Parameters | UGC (n = 51) Median [IQR] | CTC (n = 51) Median [IQR] | p-value |
|---|---|---|---|
| FluTD Time (mins) | 3 [3 – 4] | 10 [8 – 17] | <0.001** |
| Total Radiation Dose (milligray) | 1675 [1264 - 2409] | 1894 [1332 - 2668] | 0.541 |
| Total Fluoroscopy Time (mins) | 11 [9.6 - 15.7] | 14.2 [10.3 - 16.3] | 0.122 |

Abbreviations: IQR: Interquartile range, FluTD: Fluoroscopy-to-device. **Statistically significant; p < 0.05.

**Table 4. Overall procedural success, complication rates, and device efficacy, engagement, and safety parameters stratified by PCI technique.**

| Interventional Outcome Measures | UGC (n = 51) N (%) | CTC (n = 51) N (%) | p-value |
|---|---|---|---|
| Device Safety | 50 (98.04) | 51 (100) | 0.999 |
| Device Efficacy | 50 (98.04) | 50 (98.04) | 0.999 |
| Device Engagement Rate | 51 (100) | 51 (100) | 0.999 |
| Procedure Success Rate | 51 (100) | 51 (100) | 0.999 |
| Guide Catheter-induced Ostial Dissection | 1 (1.96) | 0 (0) | 0.315 |
| Use of Mother-and-Child Catheter Technique | 0 (0) | 1 (1.96) | 0.325 |
| **Change of Access Site** | | | |
| Not Required | 51 (100) | 50 (98.04) | 0.315 |
| Required | 0 (0) | 1 (1.96) | |
| **Change of Guide Catheter** | | | |
| Not Required | 50 (98.04) | 50 (98.04) | 1.000 |
| Required | 1 (1.96) | 1 (1.96) | |
| **In-Hospital Mortality** | | | |
| Absent | 50 (98.04) | 51 (100) | – |
| Present | 1 (1.96) | 0 (0) | |

to one case of coronary ostial dissection during LM intervention. Despite this complication, the device engagement rate remained at 100%. No cases of dissection occurred in the conventional group (p = 0.315). The "mother-and-child" catheter technique for device delivery was employed in one case (1.96%) in the conventional group. One instance of crossover to femoral access occurred in the CTC group due to radial spasms, and this switch was subsequently accompanied by a change in the guide catheter. In-hospital mortality was observed in one patient (1.96%) in the UGC group, and was attributed to urosepsis-pyelonephritis with multiorgan dysfunction syndrome (MODS), while no deaths occurred in the CTC group. No cases of coronary perforation, in-hospital stent thrombosis, or re-MI were observed in either cohort. Both strategies resulted in safe post-procedural outcomes, and despite a higher lesion complexity in the UGC cohort, the single catheter-based PCI strategy maintained a 100% ostial engagement rate and low complication burden. Overall, both interventional strategies demonstrated near-identical performance in terms of efficacy, engagement, procedural success, and complication rates (**Table 4**).

## Multivariate log-linear regression analysis

As evidenced in **Table 5**, multivariate log-linear regression analysis confirmed a 28.9% reduction in FluTD time in the UGC cohort (β = −1.289, P < 0.001; 95% CI: −1.446 to −1.132), relative to that of the CTC cohort. The presence of diabetes mellitus was associated with a significant increase in FluTD time by 17% (β = 0.170, p = 0.040), while age, sex, LV function, and culprit vessel involvement showed no significant association. Interestingly, nighttime procedures (Time of

**Table 5. Multivariate log-linear regression analysis of predictors of Fluoroscopy-to-Device (FluTD) time.**

| | | FluTD time | | | | |
|---|---|---|---|---|---|---|
| | | Beta-Coefficient | Std. error | p-value | 95% CI (Lower limit) | 95% CI (Upper limit) |
| Age | | −0.000 | 0.004 | 0.931 | −0.007 | 0.007 |
| Gender | Female | Reference | | | | |
| | Male | 0.048 | 0.118 | 0.687 | −0.186 | 0.282 |
| DM | Absent | Reference | | | | |
| | Present | 0.170 | 0.081 | **0.040*** | 0.008 | 0.331 |
| LV Function | | 0.009 | 0.006 | 0.125 | −0.003 | 0.021 |
| Catheter Strategy | CTC | Reference | | | | |
| | UGC | −1.289 | 0.079 | **0.001*** | 1.132 | 1.446 |
| Culprit Vessel | LAD | Reference | | | | |
| | LCX | 0.146 | 0.134 | 0.281 | −0.121 | 0.413 |
| | RCA | 0.048 | 0.101 | 0.638 | −0.154 | 0.249 |
| Time of performing PCI | Day | Reference | | | | |
| | Night | −0.157 | 0.089 | 0.081 | −0.020 | 0.335 |

Abbreviations: FluTD: first Fluoroscopy-to-device, Std. error: Standard error, DM: Diabetes mellitus, LV function: Left ventricular function, UGC: Universal guiding catheter strategy, CTC: Conventional two catheter strategy, LAD: Left anterior descending artery, LCX: Left circumflex artery, RCA: Right coronary artery, PCI: Percutaneous coronary intervention. * Statistically significant, p < 0.05

PCI) trended toward shorter FluTD time (β = −0.157; 95% CI: –0.020 to 0.335), though statistical significance was not reached (p = 0.081).

## Discussion

Contemporary percutaneous coronary intervention relies on the use of separate catheters for diagnostic angiography and subsequent angioplasty. In the present study, we investigated the safety and efficacy of a single, universal catheter against that of the dual catheter approach, in improving temporal and clinical outcomes among patients with STEMI. Compared to the conventional two catheter strategy, FluTD time was significantly lower by a margin of 7 minutes in the UGC cohort and was further corroborated by the results of a log-linear regression analysis model, which positioned the single catheter strategy as the strongest independent predictor of reduced FluTD time. Further, we found that this time reduction was correlated with improved angiographic reperfusion and reduced radiation exposure, while device engagement and procedural safety were comparable between both catheter strategies. Our findings support the notion that utilization of a universal catheter is associated with discrete benefits that were not captured in prior studies of a similar nature. To our knowledge, this is the first study to employ a procedural delay metric, distinct from total ischemic time, to compare PCI strategies and subsequently evaluate their impact on epicardial perfusion and downstream coronary microcirculation.

### Single catheter-based angiography and PCI is associated with reduced procedural duration and improved angiographic outcomes

Effective reperfusion is predicated upon clinically meaningful reduction in total ischemic time at the patient, systemic, and procedural levels [31]. Over the last two decades, concerted efforts have been directed towards analyzing and addressing delays at the patient and system level [32–37], while the focus on procedural efficiency has been limited. This, in turn, is largely due to the absence of consistently measurable and reproducible metrics that can shed light on those areas where interventional efforts need to be directed. Between arrival at the cath lab and device deployment in the culprit vessel, various patient-specific, lesion-specific, and operator/equipment-specific factors are likely to prolong the overall time to

reperfusion, but remain obscured under traditional metrics, such as D2B, that conflate multiple delays in the larger chain of reperfusion. To better focus on such delays within the cath lab, we used FluTD time as a marker of efficiency and compared the time to device activation between the UGC and CTC strategies. Our finding of reduced FluTD time with the UGC approach remains consistent with similar studies utilizing a single catheter for angiography and PCI, which have shown significant reductions in puncture-to-balloon, sheath-to-balloon, and D2B times compared to the conventional two catheter approach [16–18]. While the UGC technique represents advancements in the utilization of a true 'universal' guide catheter for both diagnostic angiography and PCI, the vast majority of studies in the literature remain focused on the use of a single catheter for angiography alone. For instance, operational use of JL as a multipurpose catheter for engagement of the left and right coronary arteries was associated with a mean reduction in procedure time by 1.2 minutes, compared to patients who underwent angiography with separate JR and JL catheters [38]. Likewise, engagement of both coronaries with a single Tiger catheter was associated with reduced contrast volume and fluoroscopy time, albeit with increased rates of catheter instability during angiographic visualization [39]. In contrast, a prospective, randomized pilot study, comparing the safety and efficacy of four universal DxTerity catheters (Medtronic, Santa Rosa, USA) against the conventional two catheter approach found no statistically significant difference in the total angiography time between the single and dual catheter groups [40]. A recent meta-analysis of seven clinical trials, incorporating data from over 2,000 patients, similarly found no difference in procedure time, fluoroscopy time, or contrast volume between the single and dual-catheter approaches for transradial coronary angiography [41]. Taken together, the utilization of single catheters for diagnostic angiography has been well studied, while their role in facilitating subsequent interventions has remained underexplored. Further, data from the aforementioned studies suggests that the durational margin between the two catheter approaches for angiographic visualization ranges between 1–2 minutes across studies, reflecting a very limited window within which clinical benefits are unlikely to arise.

Our results, therefore, are significant not only due to the relatively larger temporal gains observed as a direct result of employing the same catheter for both diagnosis and PCI, but also due to the correlation of this time reduction with clinically relevant patient outcomes. Indeed, shorter time to reperfusion has been consistently associated with reduced risk of mortality and improved LV function [11,42,43]. Within this context, we show that the reduction in time to establishment of coronary flow was accompanied by enhanced microvascular perfusion of the infarct area, as evidenced by the significantly higher proportion (86.3%) of patients who achieved a MBG of 3 in the UGC cohort (compared to 54.9% in the dual catheter group) (**Fig 1**). The direct clinical implications of this finding, however, have to be interpreted with caution. In a substudy of the APEX-AMI trial [44], shorter time to reperfusion was associated with higher MBG and improved survival, whereas no such association was observed in a substudy of the INFUSE-AMI trial [45]. Further, it is likely that the 7-minute reduction alone may not sufficiently account for the observed results, and few additional hypotheses merit consideration. With the conventional two catheter approach, operators have to use more contrast injections to obtain more angiographic visualizations from both coronary arteries, which could increase the odds of distal microembolization. Likewise, with the UGC approach, fewer catheter movements might serve to decrease the risk of thrombus fragmentation or displacement and create a protective effect on capillary patency, and might explain the differences in microvascular outcomes between our patient cohorts, despite both groups sharing a comparable burden of thrombotic lesions.

### Comparative safety and efficacy of procedural strategies in complex presentations

Our analysis of device-specific secondary endpoints revealed that the IL guiding catheter demonstrated a safety and efficacy rate of 98.04%, with a remarkable engagement rate of 100% for both coronary arteries. Regarding efficacy, a change in the guide catheter was required in one instance (1.96%), as the IL was unable to provide the backup support and force required to navigate a highly tortuous LCX. Although this finding was statistically insignificant, the requirement of an extra backup catheter might be clinically significant, considering the wealth of data that otherwise testifies to the strong backup

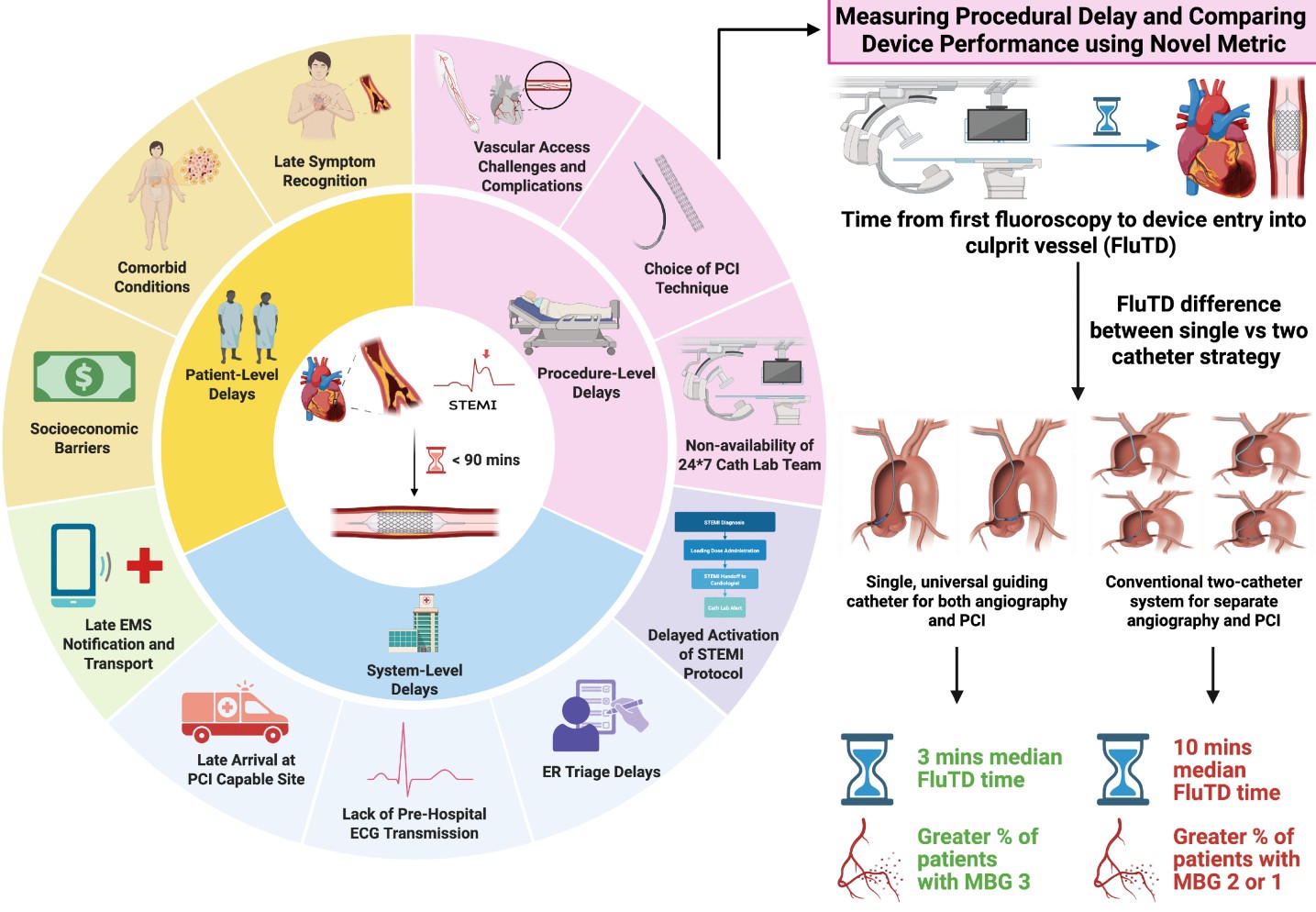

**Fig 1. Schematic representation of patient, systemic, and procedural delays in time to reperfusion and visual summary of temporal and angiographic outcomes associated with procedural optimization.**

offered by the IL [13–15]. Therefore, the performance of this catheter in vessels with marked tortuosity needs to be investigated in future studies. One incidence (1.96%) of catheter-induced ostial dissection of the left main was observed in the UGC group in a patient with pre-existing LM disease. While studies reporting the incidence of left coronary artery dissection using the IL guiding catheter are sparse, Youssef et al. reported a 0.48% incidence of RCA dissection when using the Ikari Left 3.5 guiding catheter in a prospective, single-center study [46]. However, the authors reported that this event rate dropped to 0% as operator experience increased [46], reinforcing that device-induced complications are inversely related to mastery of the learning curve. It is also worth considering that the choice of access site likely had no impact on the dissection event, as supported by prior studies [47]. Lastly, although patients in the UGC cohort presented with significantly higher baseline risk, including greater hemodynamic instability and angiographic lesion complexity, they were noted to have better angiographic outcomes. While the efficacy of the IL catheter in these specific conditions has not been investigated, it is reasonable to postulate that certain mechanistic and temporal advantages might be at play. The secondary and tertiary curve support, combined with the enhanced backup offered by the IL may facilitate more efficient wiring and catheter manipulation in bifurcations and ostial left-main lesions, leading to enhanced flow. However, the impact of procedural

optimization on microvascular perfusion outcomes among those with severe Killip classification requires careful scrutiny. It has been established that advanced heart failure correlates with impaired myocardial perfusion, independent of epicardial patency [48], and longer time to reperfusion by itself is associated with increased risk of both microvascular perfusion [28] and heart failure [49]. Against these findings, it is indeed surprising that patients in the UGC cohort achieved higher MBG, despite having a greater proportion of Killip Class II and IV presentations. While one may speculate that the 7-minute reduction in intra-procedural time might mitigate microvascular injury in critically ill patients, as they might derive disproportionate benefits from minute-level time savings due to rapidly deteriorating myocardium, it appears uncorroborated in light of the existing data. Various unmeasured confounders such as substantially shorter pre-hospital delays and operator decisions could have exerted their influence on our findings and merits exploration in future studies.

## Barriers and solutions to uptake of universal guiding catheters for percutaneous interventions

From the perspective of operators, our findings advocate for a hybrid approach that leverages the efficiency of single catheter PCI, while maintaining proficiency in bailout techniques (e.g., mother-and-child systems) for anatomically challenging cases. Institutions should integrate IL-specific simulation training to accelerate competency, especially among early adopters and create the momentum towards universal catheter use for both angiography and PCI. Despite the advantages of this technique, the second International Transradial Access Practices survey found that, among 1,065 operators around the world, nearly 63% reported using the single catheter technique solely for diagnostic angiography, while the usage of this technique for PCI was not addressed in the questionnaire [50]. Based on anecdotal evidence from a handful of operators in our country, we found that the number of cardiologists opting for a universal guiding catheter approach are few and far in between. This disparity can be attributed to several factors, including inertia in adopting new systems, a lack of awareness among physicians regarding the potential benefits of universal guide catheters, and pre-conceived notions of a long and gradual learning curve associated with mastering techniques for devices like the IL. Regarding the latter concern, we have described the engagement steps associated with the IL and offered a strategy of placing a 0.035" J Tip Teflon wire above the primary and secondary curve to better facilitate right and left coronary engagement, respectively. This simplified approach might appeal to less experienced interventional cardiologists and eventually lead to enhanced adoption of universal catheters. This shift in practice can be further motivated by the findings of a prospective, randomized study which highlighted that the single-catheter technique for coronary angiography can be performed effectively by young cardiology fellows in training, resulting in minimal complications [51]. With deliberate practice and mentor-led proctoring systems, it is possible to extend these findings to enable novice interventional cardiologists to perform both angiography and PCI with the same catheter. In such environments, it is also worth considering that FluTD time may find utility as a competency-based threshold to standardize training, ensuring procedural duration, fluoroscopy time, and complication rates mirror those of experienced interventional cardiologists.

## Temporal and economic implications of intraprocedural optimization in resource-limited settings

While the 7-minute reduction in FluTD time observed in this study might seem small in isolation, its true significance emerges when considered within the labyrinth of pre-hospital and in-hospital delays that are particularly pronounced in low and middle-income countries (LMICs). It has been established that every 30-minute delay in reperfusion increases the risk of 1-year mortality by 7.5% [52], with logistical delays influencing the time to intervention. Compared to those in high-income countries (HICs), patients living in LMICs are burdened by unique challenges right from initial symptom onset to final revascularization in the cath lab, and these compounding delays place them at a higher risk for poor long-term outcomes [53–55]. In India, for instance, pre-hospital delays remain high despite the implementation of quality improvement measures, advances in healthcare infrastructure, and gradual increases in the number of PCI-capable hospitals [53,56]. Strong public health initiatives, prompt mobilization of EMS, efficient regionalization of STEMI care, streamlined out-of-pocket and insurance payment policies, and an overall improved coordination between the pre- and in-hospital networks are crucial for

clinically relevant decreases in total ischemic time [56]. Indeed, STEMI India, a not-for-profit organization, formed a consortium with the Cardiological Society of India (CSI) and Association of Physicians of India (API) to propose a nationwide framework for ensuring timely reperfusion in patients with STEMI [57]. It is worthwhile to note that procedural delays represented a non-negligible component within this paradigm, and in high-volume LMIC cath labs, where systemic delays compound procedural bottlenecks, strategies such as single catheter-PCI may serve as a clinically modifiable lever to amplify the impact of existing public health efforts. Beyond procedural optimization, universal catheters also offer economic benefit. At our center, all guiding catheters, regardless of manufacturer, cost ₹11, 540 ($130), and the Tiger diagnostic catheter costs ₹2,810 ($32). For the UGC cohort, this amounts to a catheter cost-per-case of ₹11,540 ($130), while for the CTC cohort, the combination of a Tiger catheter plus a separate guiding catheter (JR, JL, Amplatz, or extra backup curved catheter) costs ₹14,350 ($162), yielding a per-case savings of ₹2,810 ($32) with the former technique. Although institutional purchasing and patient billing costs are likely to vary across hospitals, if one were to extrapolate the current prices to the 57, 512 primary PCIs performed in India in 2018 [58], nationwide adoption of a single-catheter strategy could conserve approximately ₹161.6 million ($1.8 million) annually. The resulting cost-per-savings holds additional significance within India's two-tiered healthcare financing landscape, where low-income earners, who makeup 50% of the population, are covered under the national public health insurance scheme, while the remaining patients pay out-of-pocket. Universal catheter adoption could, therefore, free up a significant proportion of the national budget that could be redirected towards expanded EMS coverage, ambulance transports, and other STEMI care pathways that dominate the pre-hospital components of total ischemic time. Overall, our findings suggest that utilization of a single catheter for both angiography and PCI has temporal, clinical, and economical benefits that can synergize to attenuate the degree of microvascular injury secondary to prolonged ischemic duration [59], and potentially narrow disparities in STEMI outcomes in resource constrained settings.

## Limitations and future directions

The present study is subject to all limitations inherent to retrospective analyses, including the lack of causality and control for confounding variables. Further, the small sample size and non-randomized comparison of interventions may introduce selection bias, especially for variables that were not fully adjusted for in the multivariate analysis. The study design also precluded the collection of D2B times for each patient, which hindered contextualizing FluTD time and procedural impact within the total ischemic duration. Additionally, all procedures in the UGC group were performed by a single operator, which limits generalizability of the findings across operators with varying levels of expertise. Therefore, we recognize that any observed improvement in epicardial or myocardial perfusion could be attributable to operator expertise that has been accumulated over the years of practice, in addition to any benefits conferred by the choice of PCI technique itself. When combined with the reduction in procedural duration, our data sheds light on a previously uncharacterized facet of the rapid revascularization paradigm of "time is myocardium" by suggesting that offsetting incremental delays in time to device activation might yield clinical benefits on a timescale that has conventionally remained unexplored. However, the nature and design of the study did not allow for delineation of the relative contributions of each of the study variables towards the observed patient outcomes. Although a greater proportion of patients who underwent single catheter-based angiography and PCI had better myocardial perfusion, it was only quantified using angiographic surrogates. Lastly, the impact of reduced FluTD time on clinically meaningful outcomes such as infarct size, left ventricular recovery, and long-term survival was not evaluated. To address these gaps, we plan to conduct a prospective, multi-center, randomized controlled trial aimed at better understanding the impact of procedural optimization on clinically relevant cardiovascular outcomes. This investigator-initiated trial, titled **F**luoroscopy-to-Device Time **A**ssessment of a **S**ingle, Universal Catheter vs **T**wo Catheters for **E**fficient **R**eperfusion during **P**ercutaneous **C**oronary **I**ntervention (FASTER-PCI), will evaluate the influence of FluTD time, and other metrics such as D2B and total ischemic time, on LV function and microvascular recovery via trans-thoracic echocardiogram (TTE) and cardiac magnetic resonance imaging (CMR), respectively. The trial is prospectively registered with the Clinical Trials Registry-India (CTRI) under the registration number CTRI/2025/07/090233.

## Conclusion

Our study demonstrates that a single guide catheter approach for angiography and PCI in STEMI patients significantly reduced FluTD time by 7 minutes compared to the conventional two-catheter approach, and was associated with a greater proportion of patients achieving higher myocardial blush grades. FluTD time offers a more precise and reproducible method for assessing and addressing delays within the catheterization laboratory during primary PCI in STEMI patients. Integrating this parameter into future STEMI protocols could standardize operator performance and incentivize intra-procedural optimization to minimize time to reperfusion. To this end, we also highlight the imperative for large-scale proctoring and outcome tracking to ensure technical advances translate into equitable gains in STEMI care. A simplified strategy of using a Teflon wire to ease handling and engagement of the IL guide catheter was also discussed to encourage uptake among novice interventionalists. Lastly, the universal catheter approach yielded a savings-per-case of ₹2,810 ($32), which amounts to millions of dollars at the population level that could theoretically be re-invested to curtail other bottlenecks that preclude timely reperfusion among patients living in LMICs.

## Supporting information

**S1 Table. De-identified dataset of baseline clinical, angiographic, and procedural characteristics of patients undergoing PCI with UGC or CTC strategy.**
(XLSX)

**S1 File. Authorship change request form.**
(PDF)

**S2 File. Signed letter from co-authors approving of authorship change.**
(PDF)

## Acknowledgments

The authors would like to acknowledge the timely assistance provided by the cath lab technicians – Mr. Karthi N, Mr. Macson R, Mr. Murugan P, and Mr. Jayasuriya B – in retrieving the relevant angiographic data and catheter costs. We also thank all the staff, including nurses, for ensuring smooth operation of our 24*7 high volume cath lab. Fig 1 was created using https://www.biorender.com/.

## Author contributions

**Conceptualization:** Nagendra Boopathy Senguttuvan.

**Data curation:** Mohajit Arneja.

**Formal analysis:** Mohajit Arneja, Swetharajan Gunasekar, Dharaneswari Hari Narayanan.

**Investigation:** Mohajit Arneja, Swetharajan Gunasekar, Dharaneswari Hari Narayanan, Nagendra Boopathy Senguttuvan.

**Methodology:** Mohajit Arneja, Nagendra Boopathy Senguttuvan.

**Project administration:** Joshma Joseph, Harilalith Kovvuri, Sharath Shanmugam, Vinod Kumar Balakrishnan, Jayanty Venkata Balasubramaniyan, Sadhanandham Shanmugasundaram, Sankaran Ramesh.

**Resources:** Swetharajan Gunasekar, Dharaneswari Hari Narayanan, Nagendra Boopathy Senguttuvan.

**Supervision:** Vinod Kumar Balakrishnan, Jayanty Venkata Balasubramaniyan, Sadhanandham Shanmugasundaram, Sankaran Ramesh, Nagendra Boopathy Senguttuvan.

**Validation:** Nagendra Boopathy Senguttuvan.

**Visualization:** Swetharajan Gunasekar, Dharaneswari Hari Narayanan.

**Writing – original draft:** Mohajit Arneja, Swetharajan Gunasekar, Dharaneswari Hari Narayanan.

**Writing – review & editing:** Mohajit Arneja, Swetharajan Gunasekar, Dharaneswari Hari Narayanan, Joshma Joseph, Harilalith Kovvuri, Sharath Shanmugam, Pavitraa Saravana Kumar, Asuwin Anandaram, Vinod Kumar Balakrishnan, Jayanty Venkata Balasubramaniyan, Sadhanandham Shanmugasundaram, Sankaran Ramesh, Nagendra Boopathy Senguttuvan.

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
