## [Decision Letter · Decision Letter 0]

3 Sep 2025

Dear Dr. Senguttuvan,

Thank you for submitting your manuscript to PLOS ONE. After careful consideration, we feel that it has merit but does not fully meet PLOS ONE’s publication criteria as it currently stands. Therefore, we invite you to submit a revised version of the manuscript that addresses the points raised during the review process.

Reviewer #1:

The study primarily focuses on outcomes that have already been established in the past, while myocardial blush grade (MBG) serves as a secondary outcome measure. Notably, the difference in MBG between the groups is more robust and hardly explained by time difference, which warrants further clarification and explanation.

Reviewer #2:

Thank you for giving me a chance to review the article. Overall, this is a well written article on the significance of using single universal catheter also known as IL guiding catheter in patients presenting with STEMI in a lower middle-income country. The author has critically emphasized the importance of time saving strategies like this one to improve the outcome of patients with STEMI.

I have some minor comments and queries which i am sharing for the authors response

1. Methodology: line 134-141 is a little confusing. What i understand from authors message is it is a retrospective study and an ERC exemption was obtained. But in the methodology part the author says the control arms that is CTC were enrolled prospectively? This needs clarification and if this is the case was a separate ERC obtained during that period?

Table 4: please use the term CTC to show the consistency in the manuscript. Please remove the sign> in p value 0.9

Discussion: Please reduced to six paragraphs. Use main headings as subheadings. In the current stat the discussion is too lengthy.

We look forward to receiving your revised manuscript.

Kind regards,

Sana Sadiq Sheikh

Academic Editor

PLOS ONE

Journal Requirements:

2. In the online submission form, you indicated that [The raw data underlying the results presented in the study are available from the corresponding author upon reasonable request.].

Additional Editor Comments:

Editor comments:

1. Please mention ERC number.

2. Be consistent with the terminologies. The terminologies for the catheters is different in Table 4.

3. First paragraph of discussion should summarize the key findings. Currently it has background literature. The reference to tables is not needed when summarizing the results in discussion.

4. The claim "first study" has been contraindicated by the subsequent information provided in the discussion. The authors have referred to multiple studies that have compared the intervention outcomes between two catheters.

5. Please justify the selection of patients of the UGC group from a single operator. Also, how will you address this selection bias. It is one operator vs. eight operators.

Reviewer #1:

The study primarily focuses on outcomes that have already been established in the past, while myocardial blush grade (MBG) serves as a secondary outcome measure. Notably, the difference in MBG between the groups is more robust and hardly explained by time difference, which warrants further clarification and explanation.

Reviewer #2:

Thank you for giving me a chance to review the article. Overall, this is a well written article on the significance of using single universal catheter also known as IL guiding catheter in patients presenting with STEMI in a lower middle-income country. The author has critically emphasized the importance of time saving strategies like this one to improve the outcome of patients with STEMI.

I have some minor comments and queries which i am sharing for the authors response

1. Methodology: line 134-141 is a little confusing. What i understand from authors message is it is a retrospective study and an ERC exemption was obtained. But in the methodology part the author says the control arms that is CTC were enrolled prospectively? This needs clarification and if this is the case was a separate ERC obtained during that period?

Table 4: please use the term CTC to show the consistency in the manuscript. Please remove the sign> in p value 0.9

Discussion: Please reduced to six paragraphs. Use main headings as subheadings. In the current stat the discussion is too lengthy.

Reviewers' comments:

Reviewer's Responses to Questions

**Comments to the Author**

1. Is the manuscript technically sound, and do the data support the conclusions?

Reviewer #1: Partly

Reviewer #2: Yes

2. Has the statistical analysis been performed appropriately and rigorously?

Reviewer #1: Yes

Reviewer #2: Yes

3. Have the authors made all data underlying the findings in their manuscript fully available?

Reviewer #1: Yes

Reviewer #2: Yes

4. Is the manuscript presented in an intelligible fashion and written in standard English?

Reviewer #1: Yes

Reviewer #2: Yes

Reviewer #1: The study primarily focuses on outcomes that have already been established in the past, while myocardial blush grade (MBG) serves as a secondary outcome measure. Notably, the difference in MBG between the groups is more robust and hardly explained by time difference, which warrants further clarification and explanation

Reviewer #2: Thank you for giving me a chance to review the article. Overall, this is a well written article on the significance of using single universal catheter also known as IL guiding catheter in patients presenting with STEMI in a lower middle-income country. The author has critically emphasized the importance of time saving strategies like this one to improve the outcome of patients with STEMI.

I have some minor comments and queries which i am sharing for the authors response

1. Methodology: line 134-141 is a little confusing. What i understand from authors message is it is a retrospective study and an ERC exemption was obtained. But in the methodology part the author says the control arms that is CTC were enrolled prospectively? This needs clarification and if this is the case was a separate ERC obtained during that period?

Table 4: please use the term CTC to show the consistency in the manuscript. Please remove the sign> in p value 0.9

Discussion: Please reduced to six paragraphs. Use main headings as subheadings. In the current stat the discussion is too lengthy.

**Do you want your identity to be public for this peer review?** For information about this choice, including consent withdrawal, please see our Privacy Policy

Reviewer #1: **Yes: ** Asadullah Bugti

Reviewer #2: **Yes: ** Farhala Baloch

---

## [Author Response · Author response to Decision Letter 1]

16 Sep 2025

We would like to thank the reviewers, Dr. Asadullah Bugti (Reviewer #1) and Dr. Farhala Baloch (Reviewer #2), as well as the Academic Editor, Dr. Sana Sadiq Sheikh for their valuable and insightful comments on our manuscript titled “Single Catheter Strategy for Transradial Angiography and Primary Percutaneous Coronary Intervention Enhances Procedural Efficiency, Microvascular Outcomes, and Cost-Effectiveness: Implications for STEMI Healthcare in Resource-Limited Settings”. We have carefully reviewed all the suggestions and drafted a point-by-point response below.

Reviewer Comments

Reviewer #1

1) The study primarily focuses on outcomes that have already been established in the past, while myocardial blush grade (MBG) serves as a secondary outcome measure. Notably, the difference in MBG between the groups is more robust and hardly explained by time difference, which warrants further clarification and explanation.

Author's Response: We thank the reviewer for their comment. While many outcomes included in our study have indeed been explored previously in various studies, our study explicitly aims to look at the clinical implications of procedural optimization in the context of PCI for STEMI patients. Prior studies have explored the impact of various procedural strategies on restoring epicardial flow, but few have looked at the concomitant impact on downstream coronary microcirculation. To this end, we agree with the Reviewer that the difference in myocardial blush grade (MBG) might be influenced by factors other than the 7-minute difference alone. Indeed, we included several hypotheses to explain the observed difference in the original manuscript. These clarifications can be found in Lines 391-399 in the Discussion section.

Given that it is a pilot study, we also included a detailed explanation of the limitations in the current research, where we argue that patients with better MBG could have had shorter door-to-balloon times, which aligns with evidence from the current literature, and that quantification of microvascular perfusion by angiography alone might not be sufficient. This is highlighted in Line 520-534 in the Limitations and future directions subsection of the Discussion Section in the revised manuscript.

Overall, we agree with the reviewer’s comments and would like to emphasize that we clarified the role of various confounding factors in the original manuscript and we transparently acknowledged the limitations of relying on MBG alone as a surrogate marker of microvascular perfusion. However, the results of our pilot study certainly merit further exploration, and we plan on measuring the impact of catheter strategy on microvascular perfusion by using cardiac magnetic resonance imaging (CMR) within 7 days post-PCI in a prospective, randomized trial (FASTER-PCI) that we referenced in our manuscript.

Reviewer #2

1) Methodology: line 134-141 is a little confusing. What i understand from authors message is it is a retrospective study and an ERC exemption was obtained. But in the methodology part the author says the control arms that is CTC were enrolled prospectively? This needs clarification and if this is the case was a separate ERC obtained during that period?

Author’s Response: We thank the Reviewer for their comment. We agree that the term “enrolled” could cause confusion and imply that the study was conducted prospectively. Therefore, we would like to emphasize that our study was retrospective in nature, and that we included data from 51 consecutive STEMI patients in the CTC cohort, in order to match the sample size of 51 patients in the UGC group.

We have slightly altered the wording in the manuscript to reflect this change, which can be found in Lines 137-144 in the revised manuscript and is highlighted in red.

2) Table 4: please use the term CTC to show the consistency in the manuscript. Please remove the sign> in p value 0.9

Author’s Response: Thank you for catching the oversight with respect to terminologies.

The cohort classification for Table 4 has been modified as UGC and CTC and the changes are highlighted in red in the revised manuscript. We have also removed the > sign for p-values.

3) Discussion: Please reduce to six paragraphs. Use main headings as subheadings. In the current state the discussion is too lengthy.

Author’s Response: We thank the reviewer for their suggestion. Although we agree that the discussion is lengthy, we firmly believe that each subheading and the information contained within it provides the necessary context required for satisfactory interpretation of the findings of our study. We have made deliberate efforts to contextualize each relevant finding against the literature and have included detailed insights to support our defense of the data on multiple levels. The scope of our study is evident from its title and warrants a detailed evaluation of the benefits of the single catheter strategy across procedural and economic fronts. Further, we present convincing evidence which highlights that the vast majority of operators around the world still utilize separate catheters for angiography and PCI, and the uptake of universal catheters remains low, despite the temporal advantages. Lastly, we not only emphasize the novelty of our findings, but we also clearly emphasize the limitations of the current research across multiple variables and justify the need for a prospective, randomized controlled trial. Combined with the fact that PLOS One does not mandate any word limits and instead prioritizes transparent and robust communication of our findings, we have chosen to retain the ‘Discussion’ as it currently stands.

Editor’s Comments

1) Please mention the ERC number.

Author’s Response: Thank you for the suggestion. The ethics statement and ethics committee approval number have now been included in the Methods section and have been removed from any other section in which they were originally included.

The changes can be found in Lines 108-111 in the Study Design subsection of the Methods Section and are highlighted in red font.

2) Be consistent with the terminologies. The terminologies for the catheters is different in Table 4.

Author’s Response: We thank the Editor for catching our oversight with respect to terminologies. This mistake has been corrected in the revised manuscript and the new table headings for Table 4 are highlighted in red.

3) First paragraph of discussion should summarize the key findings. Currently it has background literature. The reference to tables is not needed when summarizing the results in discussion.

Author’s Response: We thank the Editor for their suggestion. We have modified the Discussion section to include an opening paragraph that conveys the key results from our study. These changes can be seen in Lines 327-341 in the revised manuscript. We have also removed references to the tables in the discussion section.

4) The claim "first study" has been contraindicated by the subsequent information provided in the discussion. The authors have referred to multiple studies that have compared the intervention outcomes between two catheters.

Author’s Response: We would like to emphasize that our study is the first study to utilize the Fluoroscopy-to-Device (FluTD) time metric to compare the procedural efficiency of using a single guiding catheter for both diagnostic angiography and subsequent PCI versus using separate catheters for angiography and PCI, and to evaluate the impact of procedural strategy on epicardial and myocardial outcomes.

All other studies that we included in our discussion differ from the present study in the aforementioned aspects.

In Lines 358-373 in the manuscript, we refer to studies where a single catheter was used for diagnostic angiography to engage both the left and right coronary arteries. However, they switched to a different catheter for PCI. Therefore, these studies only utilized a single catheter for angiography and subsequently evaluated its impact on procedure time, fluoroscopy time, and contrast volume. In the present study, however, we use the same catheter for both angiography and PCI.

In Lines 90-96 in the Introduction and Lines 355-358 in the Discussion, we mention that prior studies in which the Ikari catheter was utilized for both angiography and PCI failed to evaluate epicardial and myocardial outcomes. These studies only measured procedure time, fluoroscopy time, sheath-to-balloon time, and radiation exposure. However, we not only examine the impact of using a single catheter for angiography and PCI on time to device activation (FluTD time), but we also evaluate the influence of catheter strategy on meaningful clinical outcomes.

5) Please justify the selection of patients of the UGC group from a single operator. Also, how will you address this selection bias. It is one operator vs. eight operators.

Author’s Response: We thank the Editor for this insightful comment. Five operators performed primary PCI using CTC strategy, of which one operator performed both CTC and UGC. We fully agree with the Editor that the UGC group suffers from selection bias. This is a key limitation of the present study, which we had acknowledged in the original manuscript. This can be found in Lines 522-527 in the Limitations and future directions subsection of the Discussion section. However, based on the results of the pilot study, we are planning on conducting a robust, prospective, multicenter randomized controlled trial to compare the procedural efficacy and efficiency of the single catheter strategy against the conventional two catheter strategy (FASTER-PCI), which we had referenced in our original manuscript. In this prospective trial, which is set to commence soon, we have included operators from various centers to mitigate operator bias and enhance the generalizability of our findings.

---

## [Decision Letter · Decision Letter 1]

14 Nov 2025

Single Catheter Strategy for Transradial Angiography and Primary Percutaneous Coronary Intervention Enhances Procedural Efficiency, Microvascular Outcomes, and Cost-Effectiveness: Implications for STEMI Healthcare in Resource-Limited Settings

PONE-D-25-37889R1

Dear Dr.. Senguttuvan,

We’re pleased to inform you that your manuscript has been judged scientifically suitable for publication and will be formally accepted for publication once it meets all outstanding technical requirements.

Kind regards,

Sana Sadiq Sheikh

Academic Editor

PLOS ONE

Additional Editor Comments (optional):

Reviewers' comments:

Reviewer's Responses to Questions

**Comments to the Author**

Reviewer #1: All comments have been addressed

Reviewer #2: All comments have been addressed

2. Is the manuscript technically sound, and do the data support the conclusions?

Reviewer #1: Yes

Reviewer #2: (No Response)

3. Has the statistical analysis been performed appropriately and rigorously?

Reviewer #1: Yes

Reviewer #2: (No Response)

4. Have the authors made all data underlying the findings in their manuscript fully available?

Reviewer #1: Yes

Reviewer #2: (No Response)

5. Is the manuscript presented in an intelligible fashion and written in standard English?

Reviewer #1: Yes

Reviewer #2: (No Response)

Reviewer #1: (No Response)

Reviewer #2: (No Response)

**Do you want your identity to be public for this peer review?** For information about this choice, including consent withdrawal, please see our Privacy Policy

Reviewer #1: **Yes: ** Dr Asadullah Bugti MBBS,FCPS,FSCAI

Reviewer #2: **Yes: ** Farhala Baloch

---

## [Editor Report · Acceptance letter]

PONE-D-25-37889R1

PLOS ONE

Dear Dr. Senguttuvan,

I'm pleased to inform you that your manuscript has been deemed suitable for publication in PLOS ONE. Congratulations! Your manuscript is now being handed over to our production team.

Kind regards,

on behalf of

Dr. Sana Sadiq Sheikh

Academic Editor

PLOS ONE